# A Review of the Mechanisms of Bacterial Colonization of the Mammal Gut

**DOI:** 10.3390/microorganisms12051026

**Published:** 2024-05-19

**Authors:** Qingjie Lin, Shiying Lin, Zitao Fan, Jing Liu, Dingcheng Ye, Pingting Guo

**Affiliations:** 1College of Animal Science, Fujian Agriculture and Forestry University, No. 15 Shangxiadian Road, Fuzhou 350002, China; lqj18259769095@163.com (Q.L.); lsy15621551956@163.com (S.L.); 18022231359@163.com (Z.F.); 2Institute of Animal Husbandry and Veterinary Medicine, Fujian Academy of Agricultural Sciences, Fuzhou 350013, China; fjliuj@163.com

**Keywords:** bacterial colonization, intestine, adhesion, receptor, probiotics, pathogens, colonization resistance

## Abstract

A healthy animal intestine hosts a diverse population of bacteria in a symbiotic relationship. These bacteria utilize nutrients in the host’s intestinal environment for growth and reproduction. In return, they assist the host in digesting and metabolizing nutrients, fortifying the intestinal barrier, defending against potential pathogens, and maintaining gut health. Bacterial colonization is a crucial aspect of this interaction between bacteria and the intestine and involves the attachment of bacteria to intestinal mucus or epithelial cells through nonspecific or specific interactions. This process primarily relies on adhesins. The binding of bacterial adhesins to host receptors is a prerequisite for the long-term colonization of bacteria and serves as the foundation for the pathogenicity of pathogenic bacteria. Intervening in the adhesion and colonization of bacteria in animal intestines may offer an effective approach to treating gastrointestinal diseases and preventing pathogenic infections. Therefore, this paper reviews the situation and mechanisms of bacterial colonization, the colonization characteristics of various bacteria, and the factors influencing bacterial colonization. The aim of this study was to serve as a reference for further research on bacteria–gut interactions and improving animal gut health.

## 1. Introduction

The gastrointestinal tract (GIT) of animals becomes colonized by a significant influx of microorganisms immediately following birth, which is collectively referred to as the intestinal microbiota. The intestinal microbiota, which includes bacteria, viruses, fungi, archaea, and protozoa, is involved in symbiotic relationships and sometimes coevolution with the GIT [1,2]. The gut flora is considered another “organ” and plays crucial roles in processes such as digestion, metabolism, immunity, and intestinal barrier functions [3,4,5]. For example, the gut flora can degrade complex dietary carbohydrates through an extensive set of enzymes that belong to the large family of carbohydrate-active enzymes and are largely absent in most mammals, yielding substantial short-chain fatty acids (SCFAs) (e.g., acetate, butyrate, propionate), lactate, or gas [6,7]. Moreover, intestinal bacteria produce a plethora of bioactive substances, including vitamins and serotonin [8], stimulate angiogenesis and enteric nerve function, protect against opportunistic pathogens, and participate in the development and regulation of the mammalian immune system [9]. However, for bacteria, it is imperative to facilitate colonization in the gut to effectively perform these functions within the GIT.

Bacterial colonization refers to bacteria originating from various environments, entering the host’s GIT, adhering to specific sites, such as mucins, epithelial cells, or the lamina propria, and continually growing and reproducing. This process serves as a crucial aspect of the interaction between bacteria and the gut, not only acting as a “bridge” for probiotic functions but also playing a pivotal role in pathogenic infections. However, bacterial colonization in the intestine involves various factors, such as the intestinal environment, bacterial adhesins, and host receptors. For example, *Escherichia coli* binds to the host receptor glycoprotein 2 (GP2) via type 1 pili adhesins or binds to the host receptor Tir by intimin for adhesive colonization [10,11,12].

Currently, there is extensive research on bacterial colonization and underlying mechanisms in the animal GIT. Therefore, this paper reviews the situation and mechanisms of bacterial colonization, the colonization characteristics of various bacteria, and the factors influencing bacterial colonization to serve as a reference for further research on bacteria–gut interactions, thus improving animal gut health.

## 2. Colonization Situation

The GIT of adult mammals hosts a substantial population of microorganisms, which can be pathogenic, parasitic, mutualistic, or commensalist [13]. Across different intestinal segments, the distribution of the gut flora is uneven, and bacterial quantities may differ among various species. For humans as mammal animals, in general, from the stomach to the duodenum, the bacterial count per gram of content ranges continuously from 10^1^ to 10^3^, followed by the jejunum and ileum, with bacterial counts per gram of content ranging from 10^4^ to 10^7^. In the colon and cecum, the bacterial count peaks, with each gram of content containing 10^11^ to 10^12^ bacterial cells [3]. The nonuniform distribution can be largely attributed to variations in the intestinal environment. In the stomach, exceptionally low pH renders the environment inhospitable for most bacteria, compelling them to migrate further downstream. Currently, several strains of *Helicobacter*, but not *Helicobacter pylori*, which are called non-*Helicobacter pylori Helicobacter* (NHPH), have been identified to colonize the stomachs of some animals, such as *Helicobacter suis* in pigs and *H. ailurogastricus, H. baculiformis*, and *H. cynogastricus* in cats and dogs, but their pathogenic significance appears to be limited [14]. The proximal small intestine (duodenum) is characterized by inhibitory factors such as bile and antimicrobial substances from the intestine, along with vigorous peristalsis, rendering it less conducive to bacterial colonization and retention. The middle and distal small intestine (jejunum and ileum) maintain a weakly acidic pH, but due to higher concentrations of oxygen and antimicrobial substances compared to those in the colon, it predominantly hosts facultative anaerobic bacteria belonging to Firmicutes and Proteobacteria [15]. In the colon and cecum, as the pH increases and the oxygen content decreases, the gut flora is primarily composed of anaerobic bacteria. For humans, these include Firmicutes (particularly the Ruminococcaceae and Lachnospiraceae families), Bacteroidetes, Actinobacteria, Proteobacteria, and Verrucomicrobia (such as *Akkermansia*) [16].

There is a protective mucus layer on the intestinal surface, especially in the colon, consisting primarily of water, electrolytes, lipids, proteins, and others [17,18]. The viscosity of the mucus layer is primarily attributed to the mucins secreted by goblet cells, with concentrations typically ranging from approximately 1% to 5% [19]. Interestingly, the mucus layer within the intestine is not uniform. Along the intestine, the proportion of goblet cells increases, resulting in increased secretion of mucin, leading to a transition from a thinner to a thicker mucus layer. Consequently, the thin mucus layer in the small intestine allows bacteria to not only adhere but also potentially navigate through it by “swimming”, thus enabling them to adhere to intestinal epithelial cells [15]. In contrast, due to the secretion of mucins by abundant goblet cells, the mucus layer in the large intestine gradually differentiates into inner and outer layers [20]. The outer layer of mucus shares similarities with that in the small intestine, while the inner layer, characterized by a smaller pore size and higher viscosity under normal physiological conditions, does not permit bacterial passage [21].

Overall, the distribution of the gut flora in the intestine is complex and multidimensional. This finding raises the following questions: what are the mechanisms by which these bacteria adhere to mucins or intestinal epithelial cells, and how do they establish enduring colonization within the intricate intestinal milieu?

## 3. Colonization Mechanisms

The primary processes of bacterial colonization include adhesion, growth, and reproduction, with adhesion serving as the prerequisite for bacterial colonization in the gut.

### 3.1. Adhesion

Adhesion is a complex and intricate process that is challenging to fully comprehend. To resist peristalsis, bacteria must either adhere to intestinal surfaces or multiply at a faster rate. Since peristaltic movement may be too rapid for significant bacterial multiplication, adherence to intestinal surfaces is an important factor contributing to the successful colonization of bacteria in the intestinal region [22]. Based on different binding modes, adhesion can be divided into nonspecific adhesion and specific adhesion.

#### 3.1.1. Nonspecific Adhesion

The temporary residence of bacteria within the intestine first requires the aid of nonspecific adhesion forces. Nonspecific adhesion means that bacteria approach the surface of target cells located through chemotaxis. It is mainly affected by molecular hydrophobicity, van der Waals forces, and electrostatic gravitational forces on the surface of bacteria and target cells, the most important of which is molecular hydrophobicity. However, the nonspecific adhesion of bacteria is reversible, and bacteria are easily dislodged by peristalsis and environmental changes in the intestine [23].

#### 3.1.2. Specific Adhesion

Specific adhesion refers to the process by which bacteria attach to mucosal receptors in the intestine via adhesins, which is a crucial step for bacteria to colonize the intestinal surface and function [24]. Bacterial adhesins are specific recognition molecules on the bacterial surface that can be categorized into fimbrial adhesins (Figure 1) and afimbrial adhesins (Figure 2).

The type 1 pili, P-pili, type IV pili, and curli in Gram-negative bacteria and several pili in Gram-positive organisms are the best-characterized pilus structures for fimbrial adhesins [28,29,30]. Type 1 pili and P-pili are categorized under chaperone-usher pili and are characterized as heteropolymeric structures. These proteins consist of one major pilus subunit protein that forms the pilus stalk and several minor subunit proteins located at the distal end [28]. Among these minor subunit proteins, FimH and PapG function as the actual adhesins for these two types of pili [31,32]. Type 1 pili, which are expressed in most members of the Enterobacteriaceae family, were reported to promote intestinal colonization and show distinct binding to epithelial cells distributed along colonic crypts in rats [33]. Type IV pili are present in both Gram-negative bacteria, such as Neisseria [34], and Gram-positive bacteria, such as *Clostridium perfringens*, demonstrating their widespread distribution across different bacterial taxa. Research has shown that the PILC protein at the tip of type IV pili serves as the adhesin for *Neisseria*’s attachment to the small intestine mucosa [34,35]. In *Clostridium perfringens*, type IV pili are primarily used for gliding but are essential for the formation of the largest biofilms [36]. Within the Enterobacteriaceae family, there are strains capable of expressing an adhesive amyloid-like protein called “curli” [37]. These pili exhibit aggregative and adhesive properties, although their host specificity appears to be unclear [37,38]. While some fimbriae exist only in specific bacterial species, some bacteria can express multiple types of fimbrial adhesins.

Afimbrial adhesins mainly include bacterial surface proteins, hemagglutinins [39], lipoteichoic acid (LTA) [24], and flagella [40,41], among others. For instance, the attachment of a variety of *Lactobacilli* to mucosal surfaces is accomplished through the interaction of OppA, a superficial bacterial protein also involved in oligopeptide internalization, and the glycosaminoglycan moiety of proteoglycans that form the epithelial cell glycocalyx [42]. Many Gram-positive bacteria utilize LTA in their cell walls, which plays a significant role in bacterial adhesion. Lipoteichoic acid is formed by covalently anchoring anionic polyphosphoryl ribitol to the cell membrane [43]. The flagellum comprises a molecular motor, a hook–basal body complex, and a rotating filament and is considered to be a specialized type III secretion system derived from a common ancestral structure [44,45]. Flagellum-driven motility is crucial for many gastrointestinal pathogens, including *Vibrio cholerae*, *Campylobacter jejuni*, and *Helicobacter pylori* [44]. In addition, the potential presence of afimbrial adhesins on structures such as capsules [46] and peritrichous fibrils [47] was demonstrated long ago. All these adhesion factors are selective and are associated with specific receptors on the host cell surface. Receptors are essential for bacterial colonization in the intestine. Mucins, which serve as receptors for bacteria on the intestinal surface, are divided into two types: gel-forming mucins, which are extremely large polymers, and transmembrane mucins, which cover the apical surface of enterocytes, some of which are present in the cell membrane [48,49]. These mucins are extracellular glycoproteins with extensive glycosylation, often reaching up to 80% [50]. Their protein structure primarily consists of a repeating sequence of proline, threonine, and serine. Extensive O-glycosylation occurs on threonine and serine residues to increase the viscosity of the mucus, which endows mucins with a “bottle brush-like” structure [49]. Over 80% of the mucin mass is composed of O-glycans, and O-glycosylation is the primary modification and glycosylation type that affects mucins [17,18]. The primary components of mucins differ in various locations, with MUC2 being the predominant component of mucin in the small and large intestines, while MUC5AC is typically the main constituent of gastric mucus [18]. Notably, bacteria may be able to penetrate the abundant polysaccharides in mucins as attachment sites [51]. The variability of polysaccharides can result in different bacterial adhesion specificities or different adhesins for attachment.

There are three other types of specific receptors on the host cell surface: the first is the lipid bilayer of the host cell membrane, which includes lipids and proteins; the second is cell surface adhesion receptors, which include integrins, calmodulin, the immunoglobulin superfamily, and selectins; and the third is the extracellular matrix (ECM), which is currently the most extensively studied receptor for bacterial adhesins. The ECM is a relatively stable structure that underlies epithelia and surrounds connective tissue cells [52]. It is a complex extracellular structural network composed of a large number of molecules produced and released by cells, including collagen, fibronectin, laminin, and elastin [53,54]. Enteropathogenic *Yersinia* species express the afimbrial adhesin YadA, which possesses a unique N-terminal amino acid sequence serving as the “uptake domain”. This domain facilitates tight binding to fibronectin bound to α5β1 integrin receptors. Deletion of this domain in YadA*_pstb_* results in the loss of adhesiveness to fibronectin and invasiveness while increasing the adhesion potential to collagen and laminin. This indicates that the afimbrial adhesin YadA of *Yersinia* species can act as a receptor for these extracellular matrix components, thereby facilitating adhesion to host cells [55]. *Ligilactobacillus ruminis*, a strict anaerobic gut autochthonous commensal, relies on sortase-dependent pili (LrpCBA) for adherence to the intestinal inner walls, exhibiting broad binding capabilities to gut collagen, fibronectin, and epithelial cells [56].

However, at present, our knowledge of bacterial adhesins and their receptors may only be the tip of the iceberg, making it challenging to fully understand their mechanisms of action. This study elucidated the main adhesins of different intestinal bacteria and their receptors (Table 1).

### 3.2. Growth and Reproduction

The intestine provides a suitable environment for bacterial growth and reproduction. After bacteria adhere to the host’s gut, they require nutrients, including carbon sources, nitrogen sources, and vitamins, for growth and reproduction. These nutrients are derived from the host’s diet, intestinal circulating metabolites [72], and even gut mucus [73]. Under conditions of abundant nutrients, bacteria begin to grow and reproduce, gradually forming a biofilm. Throughout this process, modifications in gene expression patterns result in the secretion of a significant quantity of extracellular polymeric substances, which serve as the framework for the three-dimensional structure of biofilms [74]. It allows the adhesion of individual bacteria, resulting in the formation of flat or mushroom-like microcolonies. The accumulation of numerous microcolonies thickens the biofilm, further enhancing the bacteria’s adaptability to the environment, including factors such as antibiotics, immune components, temperature, and competing microorganisms [75]. As the biofilm matures, there are channels between bacterial cells that transport nutrients, enzymes, metabolic products, and waste. Moreover, bacteria can detach from biofilms, relocate to new substrates, and subsequently reproduce and migrate, which ultimately results in the formation of new biofilms [76]. Biofilms bind bacteria together to protect the bacterial population from host immune responses and antibiotic interventions [77]. This serves to prevent the loss of valuable secretions and nutrients within the population [78].

## 4. The Colonization Characteristics of Different Bacteria

The colonization mechanisms of diverse intestinal bacteria vary due to differences in adhesion factors, adhesion forces, target cells, and other pertinent factors.

### 4.1. Pathogenic Bacteria

#### 4.1.1. *Helicobacter pylori*

*Helicobacter pylori* is a prominent gastrointestinal pathogen that exhibits remarkable resilience within the highly acidic milieu of the stomach, distinguishing it as the sole bacterium capable of enduring the hostile conditions therein. Indeed, this microorganism can colonize not only the gastric environment of humans but also various animal species [79]. The abundant presence of gastric proteases and the low pH of the environment render bacterial colonization quite challenging [80]. However, research has demonstrated that *Helicobacter pylori* can secrete significant amounts of urease on its cell surface. This urease enzyme catalyzes the hydrolysis of urea, generating ammonia and bicarbonate, which are subsequently released into the bacterial cytoplasm and the surrounding milieu, creating a neutral environment around the bacterium [80,81]. As the pH increases, the mucus layer undergoes a transition from a gel-like state to a more viscous state. This transition facilitates the passage of *Helicobacter pylori* through the mucus layer and its subsequent adhesion to gastric epithelial cells. Current research suggests that *Helicobacter pylori* achieves its adhesion through its outer membrane protein [82]. A previous review revealed that *Helicobacter pylori* possesses adhesins such as BabA, SabA, Hps60, and HpaA, which bind to Leb, Lex, sulfatides, and sialic acid, respectively [27]. It has also been reported to bind to human carcinoembryonic antigen-associated cell adhesion molecules on the surface of host cells via the virulence factor HopQ and enter host cells through the virulence factor CAG type IV secretion system, leading to the development of gastritis, gastric ulcers, and gastric cancer [83,84,85].

#### 4.1.2. *Salmonella*

The genus *Salmonella* is a group of Gram-negative bacteria that parasitize the gastrointestinal tracts of humans and animals. The genus *Salmonella* is currently primarily divided into two species, *S. enterica* and *S. bongori*. Serotypes of the *S. enterica* subspecies *enterica* (subspecies I) are the predominant pathogens associated with birds and mammals [86]. Within subspecies I, numerous *Salmonella* serotypes can infect different animal hosts [87]. Due to the diversity of *Salmonella* serotypes, this article primarily focuses on *Salmonella* Typhimurium to provide an introduction. *Salmonella* Typhimurium is a widely prevalent serovar with a broad host range. The pathogenicity of *Salmonella* Typhimurium is possibly related to its fimbriae, which can effectively adhere to host surfaces and initiate disease progression [88]. *Salmonella* Typhimurium may primarily rely on several types of bacterial appendages, including type 1 fimbriae (Fim), plasmid-encoded fimbriae, long polar (LP) fimbriae, and thin aggregative fimbriae (curli), to adhere to the intestine [89]. Like other bacteria in the Enterobacteriaceae family, *Salmonella* Typhimurium also possesses the FimH adhesin at the type 1 fimbriae tip. However, the respective proteins are not highly similar and do not exhibit the same binding specificity [90,91]. LP fimbriae expressed by the lpf operon have been found to adhere to mouse Peyer’s patch cells. These *Salmonella* fimbrial adhesins have different assembly pathways, including the molecular chaperone-adhesin (chaperone-usher, CU) pathway, the extracellular nucleation-precipitation pathway, and the type IV fimbriae-specific system, such as the type II secretion system [92,93]. It is worth noting that even if different serotypes of *Salmonella* have the same type of fimbriae, their host specificity is different [88]. Due to variations in the amino acid residues of the FimH tip of *Salmonella* fimbriae across different serotypes, the allelic genes can be replaced based on the specific serotype. Changes in amino acid residues lead to a structural transformation in the adhesin domain of FimH, despite its location outside the lectin domain, thus influencing its host specificity. For instance, a switch in two amino acids (serine and isoleucine) at position 56 of the *fimH* gene can lead to a change in host specificity, causing it to exhibit mannose or nonmannose receptors, depending on whether the host is a mammal or a bird [88,94,95].

#### 4.1.3. *Escherichia coli*

*Escherichia coli* is a commensal bacterium in the animal gut, with only a small fraction of strains causing disease under specific conditions. Pathogenic *E. coli* can be divided into six main types: enteropathogenic *E. coli* (EPEC), enterotoxin-producing *E. coli* (ETEC), enteroinvasive *E. coli* (EIEC), enterohemorrhagic *E. coli* (EHEC), and enteroaggregative *E. coli* (EAEC). Among them, EPEC was found to adhere to intestinal epithelial cells via 55- to 65-Mdalton plasmid-encoded type IV bundle fimbriae [61,96]. The intestinal epithelia of ETECs are infected by enterotoxins and colonization factors (CFs). CFs are classified into two types based on the process of CF assembly: CU pili and type IV pili, and the CU is composed of two proteins. One is the outer membrane protein usher, which convenes and coordinates chaperone-subunit complex formation into a pilus; the other is a periplasmic chaperone protein that promotes fimbrial folding, inhibits polymerization in the periplasm, and directs the protein to the usher [97]. EHEC colonization involves the induction of attaching-effacing lesions, mediated by type III secreted proteins and an outer membrane protein called intimin. ToxB, encoded on plasmid pO157, contributes to the adherence of *E. coli* O157:H7 by promoting the production and/or secretion of type III secretory proteins [98]. EAECs were found to aggregate and adhere to HEp-2 cells. At least three adhesins are associated with EAEC strains: the five variants of aggregative adherence fungal fimbriae, the aggregative forming pilus, and, more recently, a fibrillar adhesin named CS22 [99]. The highly virulent Shiga toxin-producing EAEC O104:H4 strain, previously isolated in Germany, was also found to contain a unique combination of the CTX-M-15 class of extended-spectrum β-lactamases and an enzyme known as serine protease self-transporting proteins of Enterobacteriacea, which contribute to greater intestinal adhesion and colonization [100,101].

#### 4.1.4. *Clostridioides difficile*

As a zoonotic pathogen, *Clostridioides difficile* has garnered significant attention. *Clostridioides difficile* is a Gram-positive bacterium capable of spore formation that primarily colonizes the host colon and causes intestinal infections and diarrhea. Its spread in the colon is a result of the use of extensive antibiotics, such as amoxicillin, cephalosporins, and clindamycin, which can lead to dysbiosis of the gut microbiota. Then, toxigenic *C. difficile* overgrows and secretes toxins, causing *Clostridioides difficile* infection (CDI). *Clostridioides difficile* adheres to and colonizes the intestine through various adhesion factors, including flagella and S-layer proteins [102]. The S-layer protein is composed of S-layer protein A heterodimers and more than 30 cell-wall proteins. Among these proteins, cell-wall protein 66 (Cwp66) is a vital adhesion factor of *C. difficile* that contains three domains: a signal peptide, three cell-wall binding 2 domains, and a variable domain. Furthermore, Cwp66, encoded by the *Cwp66* gene, is the second major cell surface antigen of *C. difficile* [102]. Currently, a large number of flagellar genes have been characterized in *C. difficile*. Among them, the flagellin and flagella cap genes (*flic* and *flid*) have been confirmed to be involved in adhesion to mouse mucus. *C. difficile* strains without flagella showed a 10% reduction in adhesion to the mouse cecum [103].

### 4.2. Probiotics

Probiotics not only enhance the intestinal microenvironment and preserve the equilibrium of the intestinal microecology but also modulate the immune response capacity of the organism, thus safeguarding intestinal health. Investigating their colonization mechanisms can contribute to understanding the regulation of the intestinal flora. The main emphasis here is on elucidating common probiotic strains within the GIT.

#### 4.2.1. *Lactobacillus*

*Lactobacillus,* which belongs to the Firmicutes phylum, is a common probiotic in mammals and has a very high potential to fight pathogenic infections and promote intestinal health [104]. The latest research indicates that a direct-fed microbial (DFM) liquid product based on *Lacticaseibacillus paracasei*, *Lentilactobacillus buchneri*, and *Lacticaseibacillus casei* enhances rumen and intestinal development in Holstein-Friesian calves, leading to increased gut diversity [105]. In the animal gut, the region between the duodenum and the terminus of the ileum, which is covered by a mucus layer, is the primary region colonized by *Lactobacillus* species [106]. In early research, the pili that cover *Escherichia coli* were found to exhibit lectin-like characteristics, revealing that bacterial adhesion factors interact with receptors through carbohydrate chains. Several years later, similar properties were found for *Lactobacillus*. Protease-treated *Lactobacillus* BG2FO4 exhibited reduced adhesion to mucus-secreting HT29-MTX cells [107]. The results of thin-layer chromatography revealed that *Lactobacillus* IFO3425 bound to carbohydrate chains contained either a galactosyl or glucosyl residue on the nonreducing terminal, revealing the interaction between glycolipid carbohydrate chains and *Lactobacillus* [108]. During this period, the mechanism of *Lactobacillus* adhesion mediated by lectin-like proteins was proposed and extensively investigated [109,110,111]. These findings provide evidence for the interaction of *Lactobacillus* with the complex web of carbohydrate chains in mucin.

*Lactobacillus* spp. possess many proteins involved in adhesion to the host on its bacterial surface according to previous research, and two primary localization patterns of surface proteins acting as adhesion factors in *Lactobacillus* have been described [106]. The first pattern involves cell-wall-anchored proteins covalently attached to the cell wall via sortases using an anchor sequence (LPXTG) located at the C-terminus. The second group comprises multifunctional proteins, which not only serve their primary intracellular functions but also act as adhesion factors. Within the former pattern, there is a notable focus on the mucin-binding (MUB) protein family, which consists of six Mub1 repeats and eight Mub2 repeats [112,113], and SpaCBA pili. Additionally, it was observed that SpaC exhibits lectin-like characteristics. Recently, elongation factor Tu, type I glyceraldehyde-3-phosphate dehydrogenase, small heat shock proteins, and 30S ribosomal proteins were identified as possible adhesion proteins of *Lactobacillus plantarum* HC-2 in the shrimp intestine [114]. In addition, *Lactobacillus plantarum* Dad-13 and *Lactobacillus plantarum* Mut-7 encode adhesion-related genes, such as genes encoding the fibronectin-binding protein and chaperone protein heat shock protein 3, in the rat intestine [115]. In the context of *Lactobacillus* adhesion to Caco-2 cells, the participation of adhesion-promoting factors has been identified. These factors have been found to enhance the adhesion of *Lactobacillus rhamnosus* GG, which typically exhibits poor adhesion [107].

Previous studies have also investigated the adhesion of different species of *Lactobacillus* and found that the adhesion of different species of *Lactobacillus* differed. *Lactobacillus reuteri* ZJ617 exhibited stronger adhesion than did *L. reuteri* ZJ615 due to the greater expression of cell-wall-associated glycerol-3-phosphate dehydrogenase (cw-GAPDH) [116], an enzyme that allows *Lactobacillus* to have greater cell permeability and adhesion to mucins. Tuomola et al. [117] tested the adhesion of 12 *Lactobacillus* species to Caco-2 cells and reported that *L. casei* (Fyos) was the most adherent strain, with approximately 14% adhering to cell cultures, while *L. casei* var. rhamnosus (Lactophilus) was the least adherent strain, with only a 3% adhesion rate. The results showed that the four most adhesive strains were *L. casei* (Fyos), *L. acidophilus* 1 (LC1), *L. rhamnosus* LC-705, and *Lactobacillus* GG (ATCC 53103). Similarly, Nadja et al. [118] tested the adhesion of 11 *Lactobacillus* strains to IPEC-J2 cells and reported that *L. reuteri* DSM 12246 had the highest adhesion rate (38%), followed by *L. plantarum* Q47 (24%). In addition, strong-adhesion strains inhibited the growth of weak-adhesion strains due to competitive adhesion between bacteria.

#### 4.2.2. *Bifidobacterium*

*Bifidobacterium*, a member of the Actinobacteria phylum, is a strictly anaerobic probiotic bacterium. It is ubiquitously present in the GIT, vaginal tract, and oral cavity of both humans and animals. *Bifidobacterium* species exhibit diverse physiological effects, including laxative properties, immune modulation, antitumor activity, and anti-aging benefits. However, research pertaining to the adhesion mechanism of *Bifidobacterium* lags behind that of its counterpart, *Lactobacillus*, with relatively few studies conducted in this area. In a recent study, surface adhesin proteins of *Bifidobacterium longum* BBMN68 were predicted, revealing a total of 21 genes encoding adhesin proteins. Through the overexpression of these genes, it was observed that only the overexpression of the *FimM* gene significantly increased the adhesion of BBMN68 to HT-29 and LS174T cells. Furthermore, the adhesion receptors for FimM were identified, revealing that fibronectin, fibrinogen, and mucin isolated from the porcine stomach act as adhesion receptors for the *Bifidobacterium longum* BBMN68 FimM protein [26]. In addition, homologs of FimM were found in *Bifidobacterium bifidum* 85B, *Bifidobacterium gallinarum* CACC 514, and 23 other *B. longum* strains [26]. These results suggest that FimM is an important afimbrial adhesin that is mainly present in *B. longum* strains. Similarly, the aggregation of *Bifidobacterium bifidum* A8 and DSM20456 was abolished after treatment with proteinase K, and this effect was more pronounced for strain A8 [119]. High adhesion of transaldolase (Tal) to *Bifidobacterium bifidum* A8 was subsequently detected via a mucin-binding assay, which indicated that transaldolase is an adhesion protein of *Bifidobacterium bifidum* A8 [119]. Unfortunately, there appears to be limited in-depth research regarding the adhesion mechanisms of *Bifidobacterium* in the gut. For instance, how their surface adhesion proteins interact with mucin and the extracellular matrix in the mucus layer and whether they directly adhere to intestinal epithelial cells remain largely unexplored.

#### 4.2.3. *Clostridium butyricum*

The probiotic effect of *Clostridium butyricum* on humans and animals is gradually gaining attention. However, there is little adhesion-related literature on this probiotic compared with the above probiotics. *Clostridium butyricum*, which also belongs to the Firmicutes phylum, is a probiotic that can produce butyric acid and is one of the earliest colonizers in infants [120]. It can regulate intestinal health by modulating microbial metabolites, such as SCFAs [121]. Currently, the precise colonization mechanism of *Clostridium butyricum* in the animal gut remains uncertain. Nevertheless, some studies have delved into its colonization properties. Previous research has indicated that the spores of *Clostridium butyricum* can withstand gastric acidity but do not germinate in low-pH and high-redox environments [122]. Subsequently, the oral administration of *Clostridium butyricum* to rats revealed that its primary colonization site is the colon [123]. Qi et al. [124] reported that *Clostridium butyricum* can adhere to mucin secreted by HT-29 cells and that knockdown of MUC2, FUT2, or GALNT7 significantly reduces bacterial attachment to HT-29 cells. These findings suggest that mucin glycans play a pivotal role in the adhesion of *Clostridium butyricum* to HT-29 cells. Furthermore, the adhesins of *Clostridium butyricum* and their receptors currently remain unclear. As a probiotic with significant potential for promoting animal gut health [125], in-depth research on this microorganism holds promise.

### 4.3. Other Commensal Bacteria

#### 4.3.1. *Escherichia coli* Nissle

*Escherichia coli* Nissle (EcN) is a nonpathogenic strain of *Escherichia coli* and is considered a probiotic that is beneficial for the treatment of Crohn’s disease, ulcerative colitis, and other inflammatory bowel diseases [126,127,128]. Earlier studies revealed that F1C fimbriae serve as adhesins that allow EcN to adhere to the mouse intestine and establish persistent colonization. In addition, six genes, including cyclic di-GMP phosphodiesterase (*gmp*), *hda*, *uidC*, *leuO*, a hypothetical protein-coding gene, and *cheZ*, have been confirmed to be associated with the colonization of Caco2 cells by EcN [129]. Unfortunately, the receptors for EcN have not been identified.

#### 4.3.2. *Enterococcus faecalis*

*Enterococcus faecalis* is a typical *Enterococcus* species that belongs to the Firmicutes phylum and is also called *Streptococcus faecalis*. However, it has a low homology of 9% with *Streptococcus*, so it is classified in the genus *Enterococcus*. *Enterococcus faecalis* can produce natural antibiotics and antimicrobial substances such as bacteriocins, which inhibit the growth of pathogens. The probiotic *Enterococcus faecium* has also been extensively reported as a pathogen. For example, it can adhere to and colonize bovine mammary epithelial cells, thus causing *E. faecalis* mastitis [130]. Colonization of the intestine may also induce liver carcinogenesis in patients with chronic hepatobiliary diseases [131,132]. The surface-exposed rhamnopolysaccharide EPA of *Enterococcus faecalis* has been confirmed to be a critical determinant of intestinal colonization. Additionally, the auxiliary sugar transferase EpaX, located in a variable region upstream of the *epa* locus, is capable of shaping the composition of rhamnopolysaccharide [133]. Similarly, it was found that the deletion of polynucleotide phosphorylase in *Enterococcus faecalis* leads to alterations of its cell wall and a greater capacity for cell adhesion [134]. Successful colonization of *E. faecalis* LX10 also led to a significant increase in the expression of adhesion genes (*znuA*, *lepB*, *hssA*, *adhE*, *EbpA*, and *Lap*) [135]. In addition, both the bacterial species and the host origin of the strain may affect the adhesion of *E. faecalis* to the intestinal mucosa [136].

## 5. Influential Factors in Bacterial Colonization

While bacteria can colonize the intestine, colonization efficiency can also be influenced by many factors, such as colonization resistance, genetics, and diet.

### 5.1. Colonization Resistance

Colonization resistance refers to the ability of the normal microbiota that has established themselves in specific locations to inhibit the recolonization of other bacteria [137]. A study revealed that germ-free mice monoassociated with a single *Bacteroides* species are resistant to colonization by the same, but not different, species. Researchers identified polysaccharide utilization loci involved in this process and named this gene locus the commensal colonization factor (CCF). When this gene is expressed normally, *B. fragilis* penetrates the colonic mucus and resides deep within crypt channels [138]. The CCF was found to induce species-specific saturable colonization, thereby preventing similar species from colonizing the intestine. The CCF system provides a basis for colonization resistance at the molecular level. However, in complex microbial relationships, mechanisms of colonization resistance are not limited. The secretion of antimicrobial products, nutrient competition, support for intestinal barrier integrity, and phage deployment are also involved and have been described previously [2,139,140].

Currently, the resistance of probiotics to pathogen colonization has been extensively explored. *Clostridium butyricum* CBM 588 can produce succinic acid, which enhances resistance to *Clostridioides difficile* colonization and protects the mouse colon from CDI [141]. The levels of neutrophils and antimicrobial cytokines in the lamina propria of the colon are also increased to enhance intestinal immunity [141]. *Clostridium butyricum* can ameliorate epithelial hypoxia caused by *Salmonella pneumoniae*, improve the stability of the intestinal flora, and inhibit the proliferation of *Salmonella* [142,143]. In addition, some *Lactobacillus* and *Bifidobacterium* species have been found to have the capacity to displace pathogens that were previously adhered to cells or mucus [144,145]. Valeriano et al. reported that *Lactobacillus mucosae* LM1 exhibited high adhesion, aggregation, and hydrophobicity and could also inhibit the adhesion of *Escherichia coli* K88 and *Salmonella* Typhimurium KCCM 40253 in the gut [146]. *Lactobacillus acidophilus* HN017, *Lactobacillus rhamnosus* DR20, and *Bifidobacterium lactis* DR10 reduced the number of enterotoxigenic *E. coli* and inhibited their colonization of cells in vitro [147]. *Lactobacillus rhamnosus* GG, *Lactobacillus debrueckii* M, *Lactobacillus plantarum* CS23, and *Lactobacillus plantarum* CS24.2 were also effective against the adhesion of *E. coli* and *Salmonella* typhimurium to mucin [148]. Similarly, *Lactobacillus acidophilus* Bar13, *Lactobacillus plantarum* Bar10, *Bifidobacterium longum* Bar33, and *Bifidobacterium lactis* Bar30 can effectively displace *Salmonella typhi* and *E. coli* H10407 from the Caco-2 cell layer. *Lactobacillus acidophilus* Bar13 and *Bifidobacterium longum* Bar33 can also modulate the immune activity of HT29 cells to generate IL-8, thereby defending against pathogen infection. The preceding findings indicate that certain intestinal bacteria can exhibit a degree of colonization resistance, particularly the resistance of certain probiotics against pathogenic bacteria, which is certainly beneficial for our means of using probiotics to avoid pathogenic infection.

### 5.2. Genetic

Genetics can also influence the colonization of the intestinal flora to a certain extent. By comparing the fecal flora of chickens, pigs, cattle, geese, and so on, it was found that the feces from the same animal had relatively high similarity [149]. Through the analysis of 5088 bacterial 16S rRNA gene sequences in the cecal microbiota of mice, it was found that the composition of the microbial community was inherited from mothers [150]. Shauni et al. employed a powerful genetic mapping approach in hybrid mice and uniquely conducted their analysis on microbial traits measured at the gut–mucosal interface. In a total of 153 core taxonomic groups, they identified 21 DNA taxa and 30 RNA taxonomic units, all displaying significant heritability estimates ranging from 39% to 83% [151]. Similarly, host genetic studies on male Hu sheep revealed heritability of diversity within the rumen microbiota, marking the first investigation of heritability of rumen microbial communities in sheep [152]. Furthermore, by examining microorganisms in the feces of 1126 twin pairs, 8.8% of all intestinal flora were found to be heritable, including Actinobacteria and Peptococcaceae spp. [153]. The above results suggest that the intestinal microbiota composition is influenced by genetic inheritance. However, the extent to which this high heritability is influenced by shared genotypes versus shared environments remains unclear. Under controlled conditions that manage dietary, age, and socioecological variations among wild baboons, 97% of the microbiome phenotypes exhibit significant heritability. Nevertheless, as host age and environmental shifts occur, the heritability estimates of gut microbiome composition may vary over time [154].

### 5.3. Diet

Diet serves as the most immediate connection between the GIT and the external environment. Dietary substrates are key determinants of the structure and function of the GIT microbiome, influencing the production of metabolites and microbe–host interactions [155]. Currently, the relationship between diet and gut microbiota has been extensively studied and reviewed [8,156]. Recent research has utilized isotope tracking of bacteria-specific protein sequences to determine the nutritional preferences of the gut microbiota in vivo [157]. By tracking 13C and 15N in the mouse gut microbiome, it was found that the primary input for microbial carbohydrate fermentation is dietary fiber, while dietary proteins are the main inputs for branched-chain fatty acids and aromatic metabolites.

Dietary fiber, which is difficult for animals to digest, serves as an essential energy source for gut bacteria. Dietary fiber provides a plethora of substrates for fermentation reactions carried out by specific species of microbes (e.g., *Bifidobacterium*, *Faecalibacterium*) that express adequate enzymatic machinery to break down these complex carbohydrates and then produce SCFAs [158]. When dietary fiber intake is insufficient, certain gut bacteria can be induced to utilize glycosylated mucin as an alternative source of nutrients, which can have deleterious consequences on the intestinal mucus barrier, consequently increasing the risk of pathogen infections [73,158]. For example, compared with grain-based chow in mice, a Western-style diet, which is high in fat and low in fiber, altered the dynamics of *Citrobacter* infection and reduced initial colonization and inflammation but frequently resulted in a persistent infection that was associated with low-grade inflammation and insulin resistance [159]. Therefore, adequate dietary fiber intake is beneficial for maintaining mucous layer integrity, supporting the normal colonization of commensal bacteria, and promoting gut health. Recent research indicates that chronic alcohol consumption leads to dysbiosis in the gut microbiota of mice, facilitating colonization of *Klebsiella pneumoniae* in the intestinal tract. This is attributed to the inhibitory effect of high concentrations of secondary bile acids, particularly deoxycholic acid, on the proliferation of *Klebsiella pneumoniae* in the gut. However, alcohol-induced dysbiosis results in reduced levels of secondary bile acids, leading to the loss of inhibition against *Klebsiella pneumoniae* [160]. In another study, feeding mice with Autoinducer-2 following antibiotic treatment-induced dysbiosis promoted the colonization and biofilm formation of *Lactobacillus rhamnosus* GG, thereby ameliorating gut dysbiosis in mice [161].

The different protein compositions in the diet also have an impact on the colonization composition of intestinal bacteria. Ortman studied the influence of two high-quality protein sources (microbially enhanced soybean meal (MSBM) and fish meal (FM)) on the colonic bacteria of weaned piglets. The research revealed that while the protein source did not impact the overall diversity of the bacterial groups, it could enhance the presence of specific members of the *Lactobacillus* genus. For instance, the abundance of *Lactobacillus delbrueckii* was highest in the MSBM group at 11.3%, compared to 3.3% in the FM group [162]. Conversely, dietary supplementation with glutamine did not affect the ratio of Firmicutes to Bacteroidetes in the jejunum of mice but could alter the ratio of Firmicutes to Bacteroidetes in the ileum [163]. In addition, when pectin and soybean meal were added to the piglets’ diets, an increase in the relative abundance of *Proteus* spp. and a decrease in the proportion of *Lactobacillus* were found [164].

In summary, diet exerts direct or indirect influences on the establishment of intestinal bacterial colonization. Currently, there is a growing research emphasis on manipulating dietary components to correct imbalances in the gut microbiota.

## 6. Conclusions

Currently, through advanced molecular biology and genomics technologies, we have gained a deeper understanding of the diversity and functions of gut bacteria. The study of bacterial colonization mechanisms plays a crucial guiding role in understanding the principles of bacterial interactions with the intestinal tract. First, in-depth research into colonization mechanisms is expected to unveil the complex interactions between the gut microbiota and the host immune system. In particular, in recent years, research on the cross-talk between the gut microbiota and intestinal stem cell niche has emerged as a new area for maintaining intestinal homeostasis [165]. This suggests that intestinal stem cells and their relationship with the intestinal microbiota provide a feasible pathway for alleviating intestinal diseases. This knowledge will contribute to understanding how microbes regulate the host immune response, laying the foundation for developing more precise regulatory methods to maintain immune balance and prevent the occurrence of diseases. Second, research on gut bacterial colonization will increasingly involve the field of metabolic regulation. A profound understanding of the interplay between microbes and host metabolism holds the potential to provide new avenues for the prevention and treatment of metabolic diseases. However, current research on the colonization mechanisms of pathogenic bacteria appears to be more extensive than that of probiotic bacteria. Research on probiotics has focused primarily on their ability to compete with pathogenic bacteria and has overlooked their own colonization mechanisms, resulting in our limited understanding of the intestinal colonization processes of various probiotics. Therefore, there is an urgent need to broaden our exploration of probiotic colonization mechanisms. For example, figuring out how probiotics find their way to receptors in the animal gut, what molecules are involved in settling in, and details such as when and where they settle helps us use probiotics more effectively. To accomplish these tasks, perhaps more feeding trials would be necessary to achieve the desired outcomes. In summary, a better understanding of the mechanisms governing bacterial colonization in the animal GIT could signify a substantial stride forward in the maintenance and promotion of animal intestinal health.

## Figures and Tables

**Figure 1 microorganisms-12-01026-f001:**
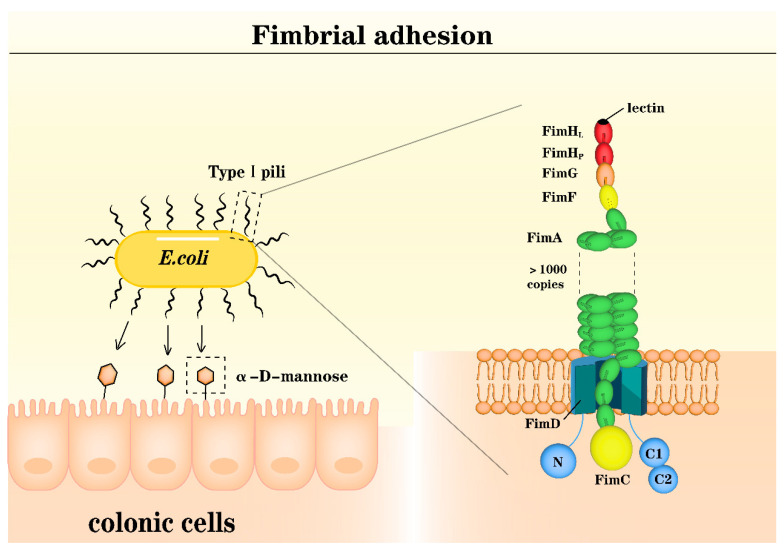
Fimbrial adhesions. Like type 1 pili in *Escherichia coli*, fimbriae are mainly composed of various Fim protein subunits, including FimH, FimG, FimF, FimC, and thousands of FimA. The FimH subunit, located at the top, utilizes binding sites (lectins) to bind with α-D-mannose-specific receptors [25].

**Figure 2 microorganisms-12-01026-f002:**
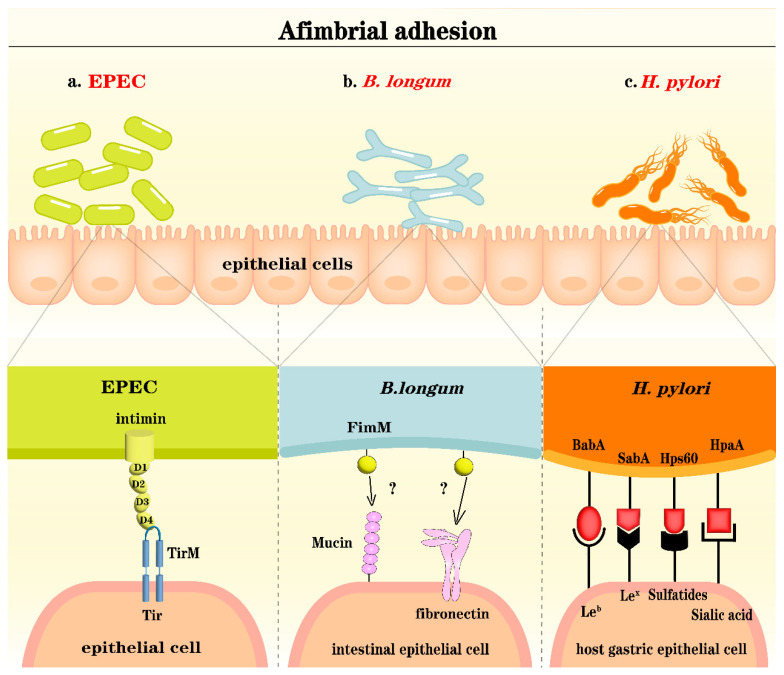
Afimbrial adhesions. The extracellular carboxy terminus of the enteropathogenic *Escherichia coli* (EPEC) protein, which is involved in afimbrial adhesion, contains three immunoglobulin domains (D1, D2, and D3) and a terminal C-type lectin cell-binding domain (D4). Intimin binds to the extracellular intimin-binding domain of Tir, Tir-M, to produce the characteristic intimate EPEC adhesion, actin accretion, and attaching/effacing lesion formation (**a**) [11]. *Lactobacillus longum* adheres to intestinal epithelial cells through the afimbrial adhesin FimM, which binds to mucin and fibronectin adhesion receptors on the cell surface (**b**) [26]. In *Helicobacter pylori*, BabA binds to mucins decorated with Leb blood group antigens. SabA binds to both Lex-sialylated and asialylated Laminin. HpaA can bind to sialic acid, and Hsp60 can bind to sulfatides (**c**) [27]. The bacterial–host adhesion models mentioned above were derived from the respective cited references and have been modified accordingly.

**Table 1 microorganisms-12-01026-t001:** Main adhesins and receptors.

Type	Host Receptor	Bacteria	References
Fimbrial adhesins			
Type 1 pili	Glycoprotein 2 (GP2) α-D-mannose	*Escherichia coli*	[10,25]
K88	Mucin-type Sialoglycoproteins	*Escherichia coli*	[57,58]
Type Ⅰ pili		*Enterobacter cloacae*	[59]
Type Ⅰ pili		*Salmonella enterica* serovar Enteritidis	[60]
Type IV pili		*Escherichia coli* EPEC	[61]
Afimbrial adhesins			
BtaE		*Brucella suis*	[62]
Intimin	Tir	*Escherichia coli*	[11,12]
TibA		*Escherichia coli*	
		*Escherichia coli* ETEC	
Collagen binding protein, CpCna		*Clostridium perfringens*	[63]
PagN protein		*Salmonella enterica* serovar Typhimurium	[64]
SD-repeat protein G	Fibrinogen	*Staphylococcus epidermidis*	[65]
Collagen binding protein, CpCna		*Staphylococcus aureus*	[66]
Yersinia adhesin A (YadA),		*Yersinia enterocolitica*	[67]
Cell-wall proteins CD2787 and CD0237		*Clostridioides difficile*	[68]
Mannose-specific adhesin	Mannose	*Lactobacillus plantarum*	[69]
High-molecular-mass cell surface protein		*Lactobacillus reuteri*	[70]
Methionine sulfoxide reductase			
Lipoteichoic acid		*Lactobacillus johnsonii*	[71]
FimM	Fibronectin/fibrinogen/mucin	*Bifidobacterium longum*	[26]
BabA, SabA, Hps60, and HpaA	Leb, Lex, sulfatides, and sialic acid	*Helicobacter pylori*	[27]

## Data Availability

Not applicable.

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
