# Peer review of "A Review of the Mechanisms of Bacterial Colonization of the Mammal Gut"

_microorganisms, 2024, doi:10.3390/microorganisms12051026_

Round 1

Reviewer 1 Report

Comments and Suggestions for Authors

This is a very comprehensive review tackling mechanisms of bacterial colonization, the colonization characteristics of various bacteria, and the factors influencing bacterial colonization in the animal intestines. Only minor revisions are needed.

Please adhere to the most recent and valid microbial nomenclature. For example, Clostridium difficile is known as Clostridioides difficile for some time now. Also, the authors sometimes italicize phyla and sometimes not, which should be unified (of note, phyla are not italicized). Salmonella serovars should not be italicized.

When influential factors in bacterial colonization, a part on genetics (section 5.2.) should be expanded significantly; there is a lot of newer literature on this in the field. When specific strains of Helicobacter pylori are mentioned (lines 67-68), more specifics should be provided.

No studies on the topic from 2024 have been cited, although the authors state that this is currently a budding field of research. For example, the authors should refer to the work of Alawneh et al. from 2024 (Animals 2024, 14, 693; https://doi.org/10.3390/ani14050693) and find similar studies to corroborate certain claims in the manuscript. In the conclusion, the authors state that there is a need to broaden the exploration of probiotic colonization mechanisms, but exact approaches should be suggested.

Comments on the Quality of English Language

Minor amendments are recommended for clarity purposes.

Reviewer 2 Report

Comments and Suggestions for Authors

This review of bacterial colonization in the mammal gut is thorough, covering many references. The questions being covered are interesting, and the figures excellent. The structure of the paper from bacterial adhesion in general to specific microbes (pathogens and probiotics separately) is a good idea, and the material is interesting even to non-specialists.

Specific science, grammar, and stylistic comments:

Title - The title is too broad, while the paper is more focused. I recommend: "A review of the mechanisms of bacterial colonization of the mammal gut."
9 - The bacteria in a gut need not have co-evolved with their host at all. They could be picked up from the environment. Even organisms that depend on gut micorbes for survival do not necessarily have co-evolved with microbes if the roles can be filled by a large number of cosmopolitan microbes. I would thus delete "that have coevolved with their hosts"
30 add "sometimes coevolution"
31 Replace "Among these, the gut flora, particularly bacteria, which are the predominant components of the animal gut microbiota, are" with "The gut flora are" - as "gut flora" and the functions thereof are not limited to bacteria
47 comma needed before "but also"
59 Parasitism is technically a kind of symbiosis. Gut microbes can be pathogenic, parasitic, mutualistic, or commensalist.
61-64 is this general for all mammals, from humans to cows to rabbits? What is the evidence from paper 3?
75-79 Same here. Are these general statements about mammal guts, or values for a specific organism?
99 Delete "Based on current research, it is evident that"
112 delete "and are"
118+175 These sub-sub-headings can probably be deleted.
266+483 lowercase t needed for typhimurium
487 forgot to italicize plantarum
488 is there a difference between Salmonella enterica serovar Typhimurium and Salmonella typhimurium?
526 "chow" is too informal if you are describing human food. You can say "diet" here, unless this is an animal study. In either case, state what animal this study (155) looked at.
